# Quantifying the contribution of *Plasmodium falciparum* malaria to febrile illness amongst African children

Ursula Dalrymple[1,2]*, Ewan Cameron[2], Samir Bhatt[2,3], Daniel J Weiss[2], Sunetra Gupta[1], Peter W Gething[2]*

[1]Department of Zoology, University of Oxford, Oxford, United Kingdom; [2]Big Data Institute, Li Ka Shing Centre for Health Information and Discovery, University of Oxford, Oxford, United Kingdom; [3]Department of Infectious Disease Epidemiology, Imperial College London, London, United Kingdom

**Abstract** Suspected malaria cases in Africa increasingly receive a rapid diagnostic test (RDT) before antimalarials are prescribed. While this ensures efficient use of resources to clear parasites, the underlying cause of the individual's fever remains unknown due to potential coinfection with a non-malarial febrile illness. Widespread use of RDTs does not necessarily prevent over-estimation of clinical malaria cases or sub-optimal case management of febrile patients. We present a new approach that allows inference of the spatiotemporal prevalence of both *Plasmodium falciparum* malaria-attributable and non-malarial fever in sub-Saharan African children from 2006 to 2014. We estimate that 35.7% of all self-reported fevers were accompanied by a malaria infection in 2014, but that only 28.0% of those (10.0% of all fevers) were causally attributable to malaria. Most fevers among malaria-positive children are therefore caused by non-malaria illnesses. This refined understanding can help improve interpretation of the burden of febrile illness and shape policy on fever case management.

DOI: https://doi.org/10.7554/eLife.29198.001

**\*For correspondence:**
ursula.dalrymple@zoo.ox.ac.uk (UD);
peter.gething@bdi.ox.ac.uk (PWG)

**Competing interests:** The authors declare that no competing interests exist.

## Introduction

Following new case management guidelines issued by the World Health Organization in 2010, individuals presenting with fever at a health clinic in sub-Saharan Africa have an increased chance of receiving a rapid diagnostic test (RDT) prior to receiving antimalarial treatment (*World Health Organization, 2015*). This reduces the over-prescription of antimalarial drugs to uninfected patients while ensuring patent infections are identified and the parasites cleared through treatment. However, in highly endemic areas, malaria infections are both common and frequently asymptomatic (*Bousema et al., 2014*), meaning that for many patients presenting at health care facilities, their fever and parasitaemia are not causally related.

Ambiguity around causality in RDT-positive fever cases is problematic from both the disease surveillance and health system perspectives. Routine case surveillance systems generally report the incidence of clinical malaria based on counts of febrile individuals with a presumed or confirmed malaria infection (*World Health Organization, 2016*). If the malaria infections in many of these cases are in fact asymptomatic then the resulting case reports will over-represent clinical malaria, even where RDT testing is ubiquitous, leading to an overestimation of morbidity due to malaria. Conversely, fevers caused by another pathogen but coincident with a malaria infection will suffer a systematic under-reporting under this protocol and consequently the disease burden of non-malarial febrile illness (NMFI) will be underestimated. In instances where a febrile individual has both a malaria infection and a NMFI, both infections can simultaneously be the cause of the fever. Indeed co-infections

have been known to modulate both the parasite load of both diseases and the severity of the individual's fever (*French et al., 2001*; *Hartgers and Yazdanbakhsh, 2006*).

Diagnosing and treating NMFI is challenging as symptoms of many of these diseases can be non-specific and similar to malaria, for example bacterial infections such as pneumonia (*Hildenwall et al., 2016*; *Källander et al., 2004*) and meningitis (*Gwer et al., 2007*). Additionally, routine diagnostic tests have not yet been developed or are not commonplace for many of these diseases (*Chappuis et al., 2013*). From the health systems perspective, case management may be inadequate if individuals who receive a positive RDT result but have a co-infection with another pathogen do not receive effective treatment for their fever. For these reasons, there is a pressing need to understand the contribution of malaria to febrile illness and how this varies spatially and temporally across endemic Africa.

### Estimating malaria-attributable fever prevalence

The rate of fever associated with (but not necessarily caused by) a malaria infection has been shown to have declined over time in some settings, halving between 1986 and 2007 in one collation of field studies (*D'Acremont et al., 2010*); a trend likely driven by the declining prevalence of malaria over the latter portion of that time period. Estimating the fraction of these fevers in which the malaria infection is the causal pathogen (i.e. the malaria-attributable fraction) is challenging, and various methods have been developed. Attempts have been made to measure the malaria-attributable fraction arithmetically using case-control trials, where the difference in the rate of malaria positivity in febrile and afebrile individuals is used to calculate the attributable fraction (*Ehrhardt et al., 2006*; *Schellenberg et al., 1994*; *Smith et al., 1994*). Alternatively, logistic regression approaches have been developed to estimate causal fractions based on measurements of blood parasite density. This approach has been demonstrated in numerous field trials, including various settings in Kenya (*Afrane et al., 2014*; *Bloland et al., 1999*), in an area of seasonal transmission in Burkina Faso (*Bisoffi et al., 2010*), and in a national survey in Mozambique (*Mabunda et al., 2009*). Computational simulations of malaria transmission can also be adapted to estimate an upper bound on the malaria-attributable fraction of fevers at varying levels of *Plasmodium falciparum* prevalence (*Pf*PR), by monitoring the proportion of parasite-positive individuals within a simulation with a symptomatic infection (here, a symptomatic infection is synonymous with a malaria attributable fever, as co-infections with other pathogens are not typically simulated) (*Griffin et al., 2010*; *Ross et al., 2006*). None of these approaches make use of the rich trove of national household survey data recording malaria infection status as well as fever history that are now available for multiple countries and years from sources such as the DHS Program (*DHS Program, 2017*) and the UNICEF Multiple Indicator Cluster Surveys (*UNICEF, 2017*).

### Using household survey data to model malaria-attributable fever and non-malarial febrile illness

Household survey data on malaria infection status and two-week fever history do not allow direct attribution of malaria causality at the individual level. However, when data from multiple individuals are combined, then the causal contribution of malaria infections to fevers within the group can be explored by measuring the extent to which fevers are more common in infected versus uninfected individuals. Building on this intuitive logic, we have developed a multinomial geospatial model (described in detail in Materials and methods) that uses the georeferenced survey data on individual-level malaria infection and fever status to infer the community-level fraction of malaria positive fevers that are either caused by or coincident with underlying malaria infections, and subsequently to map these quantities across sub-Saharan Africa. This refined understanding of the contribution of malaria versus other causes to febrile illness can improve disease burden estimation and interpretation and thus inform case management policy in sub-Saharan Africa.

## Results

### Model overview

Using a total of 38 household surveys in 24 Saharan African countries collected between 2006 and 2014, we collated 155,369 observations of two-week fever prevalence and RDT diagnostic outcome

for *P. falciparum* in children from 10,606 locations with two modelled predictor variables: *P. falciparum* prevalence in children under five years of age (*Pf*PR$_{0-5}$); and suitability for fever without a malaria infection. The final hierarchical Bayesian model predicted the proportion of individuals with a fever directly attributable to malaria (hereafter Malaria Attributable Fever, MAF) and the proportion of individuals with a fever not attributable to malaria (hereafter Non-Malarial Febrile Illness, NMFI) within each 5 × 5 km pixel across the stable *P. falciparum* transmission zones on the African continent for each year 2006–2014. These maps were used to derive a series of further metrics as outlined below. Details of model validation are provided in Materials and methods and *Figure 1—figure supplement 1*.

## Prevalence of all-cause fever

All-cause fever prevalence was calculated as the sum of the two metrics estimated by the model: prevalence of *P. falciparum* malaria-attributable fever (MAF) and non-malarial febrile illness (NMFI). In 2014, the prevalence of fever of any cause across the stable limits of *P. falciparum* transmission was 31.0%, and increased slightly over the study period, from 27.0% in 2006. Countries were highly heterogeneous in their overall fever burden, with Niger (53.4%), Gabon (44.9%) and Nigeria (42.8%) having the highest all-cause fever prevalence in 2014. The countries with the lowest fever burden in the same year were Swaziland (4.60%), Eritrea (5.3%) and Somalia (10.1%) The mapped posterior prediction of all-cause fever prevalence across Africa is shown in *Figure 1*.

## Malaria-attributable fevers among malaria-infected children

In 2014, we estimate that less than one-third (28.0%) of all fevers in *P. falciparum* malaria-infected children under five were attributable to *P. falciparum* in any given two-week period. This fraction varied geographically, as shown in *Figure 2*, with the largest contribution of MAF to malaria-positive fevers in 2014 in Swaziland (86.9%), Eritrea (81.9%) and Somalia (69.9%) and the lowest in Niger (14.7%), Gabon (20.9%) and Nigeria (20.9%). The fraction also varied though time, decreasing continent-wide from 36.1% in 2006 to 28.0% in 2014. Time-series estimates for all countries are shown in *Figure 3*.

Age of the child was also shown to have an effect on the likelihood of a child developing a malaria-attributable fever. *Figure 4* shows the fitted relationship between local *Pf*PR$_{0-5}$ and the probability of a malaria-attributable fever in the past two weeks, for all children under five years of age in *Figure 4a* and in individuals over and under two years of age in *Figure 4b*. For all children, the probability of an attributable fever grew with *Pf*PR$_{0-5}$, with a decreasing gradient as *Pf*PR$_{0-5}$ increases. *Figure 4b* shows that this trend is also true for children over two years of age, but the probability of a malaria-attributable fever increases linearly with *Pf*PR$_{0-5}$ for children under two years of age.

## Malaria infections within all fevers

In 2014, we estimate that 35.7% of all fevers in children under five were accompanied by an RDT-patent *P. falciparum* infection, and that this fraction has declined from 48.5% in 2006. The countries with the highest proportion of malaria-positive fevers in 2014 were the Central African Republic (71.0%), Equatorial Guinea (66.2%) and Guinea (65.7%) and the lowest were Ethiopia (1.4%), Botswana (1.6%) and Swaziland (2.9%). *Figure 5* highlights the large disparity between the fraction of all fevers that are malaria positive (35.7% continent-wide in 2014) and the fraction of all fevers that are causally attributable to malaria (10.0% continent-wide in 2014).

## Contribution of non-malarial febrile illness to all fevers

Non-malarial febrile illness (NMFI) arises in two forms: (i) fevers in malaria-negative individuals, and (ii) fevers in *P. falciparum* malaria-positive individuals where the fever is coincident with but not caused by the malaria infection (hereafter referred to as Malaria-Coincident Fevers, MCF). The sum of these two types of febrile illness are referred to here as NMFI, and are estimated directly by the model. The fraction of all fevers that are due to NMFI increased over the study period, from 82.5% in 2006 to 90.0% in 2014. The countries with the largest fractions of NMFI to all fevers were Ethiopia (99.4%), Botswana (99.2%) and Gambia (98.5%), and the smallest were Guinea (58.2%), Equatorial Guinea (60.4%) and Central African Republic (64.6%).

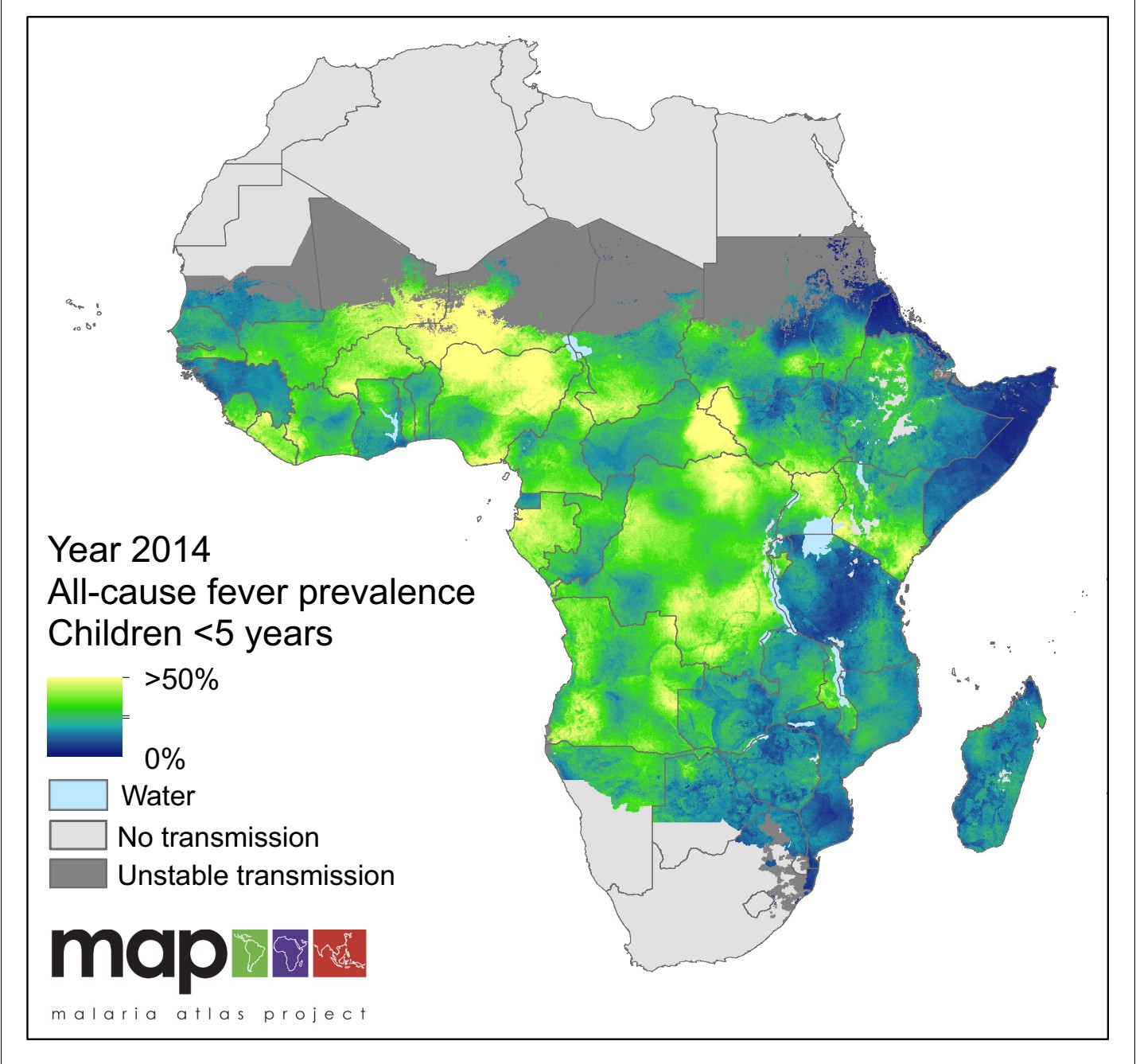

**Figure 1.** Predicted all-cause fever prevalence within limits of stable *P. falciparum* transmission in children under 5 years of age in 2014.
DOI: https://doi.org/10.7554/eLife.29198.002

The following figure supplement is available for figure 1:

**Figure supplement 1.** The histogram of the probability integral transform diagnostic.
DOI: https://doi.org/10.7554/eLife.29198.003

The proportion of MCF within NMFI fell over the study period, from 37.6% in 2006, to 28.6% in 2014. The countries that had the highest prevalence of MCF within NMFI were the Central African Republic (55.2%), Burkina Faso (54.9%) and Equatorial Guinea (44.2%), and the countries with the lowest prevalence of MCF within NMFI were Swaziland (0.4%), Botswana (0.8%) and Ethiopia (0.8%). A corollary of this is that Swaziland, Botswana and Ethiopia were also the countries with the lowest

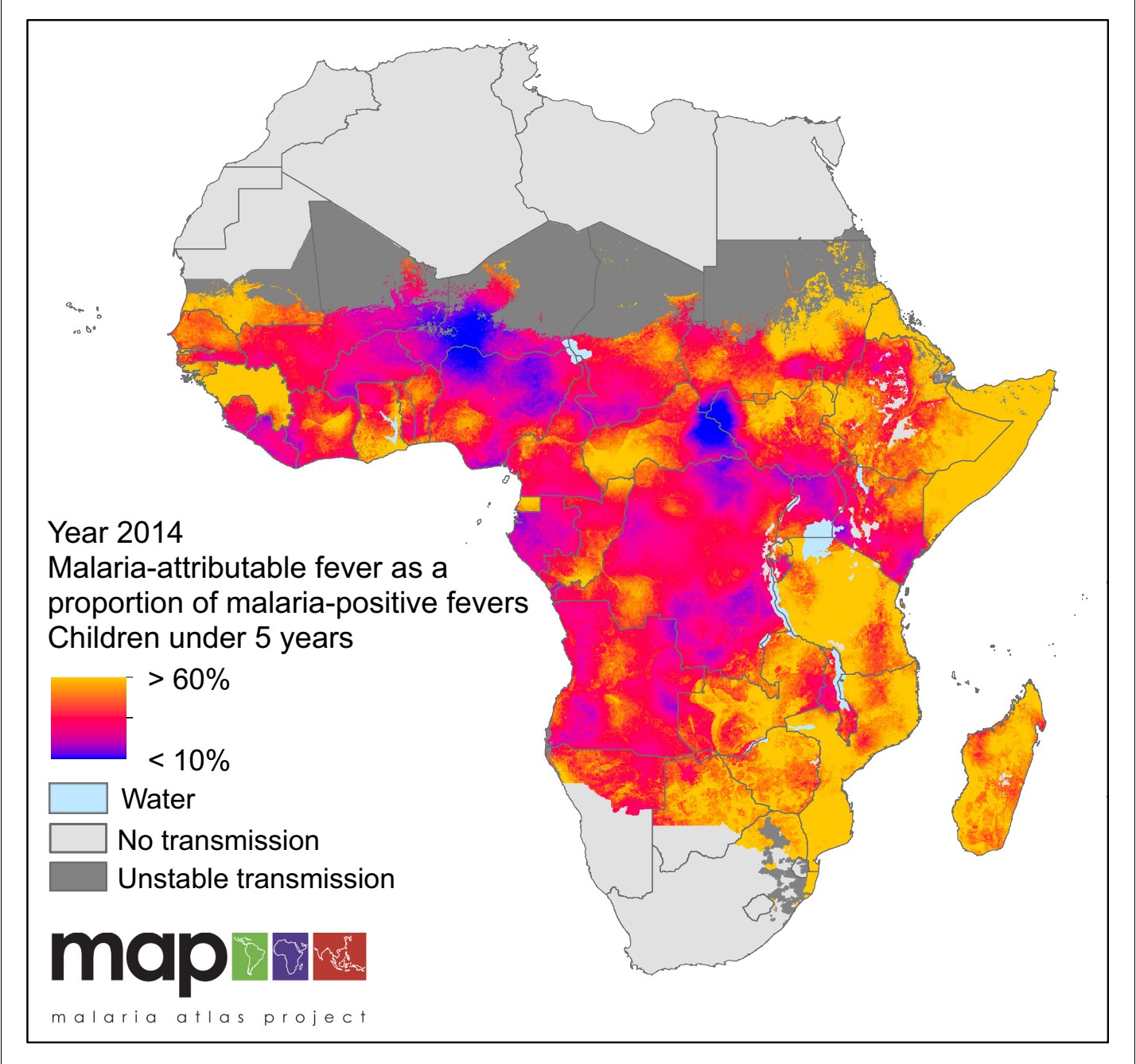

Year 2014
Malaria-attributable fever as a
proportion of malaria-positive fevers
Children under 5 years

> 60%

< 10%

Water

No transmission

Unstable transmission

**Figure 2.** Predicted malaria-attributable fevers as a proportion of malaria-positive fevers (children under 5 years of age, 2014). Predictions are shown within the limits of stable *P. falciparum* transmission.

DOI: https://doi.org/10.7554/eLife.29198.004

proportion of malaria-positive individuals within the national cases of NMFI. The posterior estimation of prevalence of NMFI in children less than five years of age within the spatial limits of *P. falciparum* transmission is shown in *Figure 6*.

Full details national-level estimates of MAF, MCF, malaria-positive fevers and NMFI can be found in *Supplementary file 1*. A plot detailing the relationship between all-cause fever, malaria-positive fevers, and MAF with varying *P. falciparum* transmission intensity in both response data and predictions can be found in *Figure 5—figure supplement 1*. These plots of our estimates are displayed

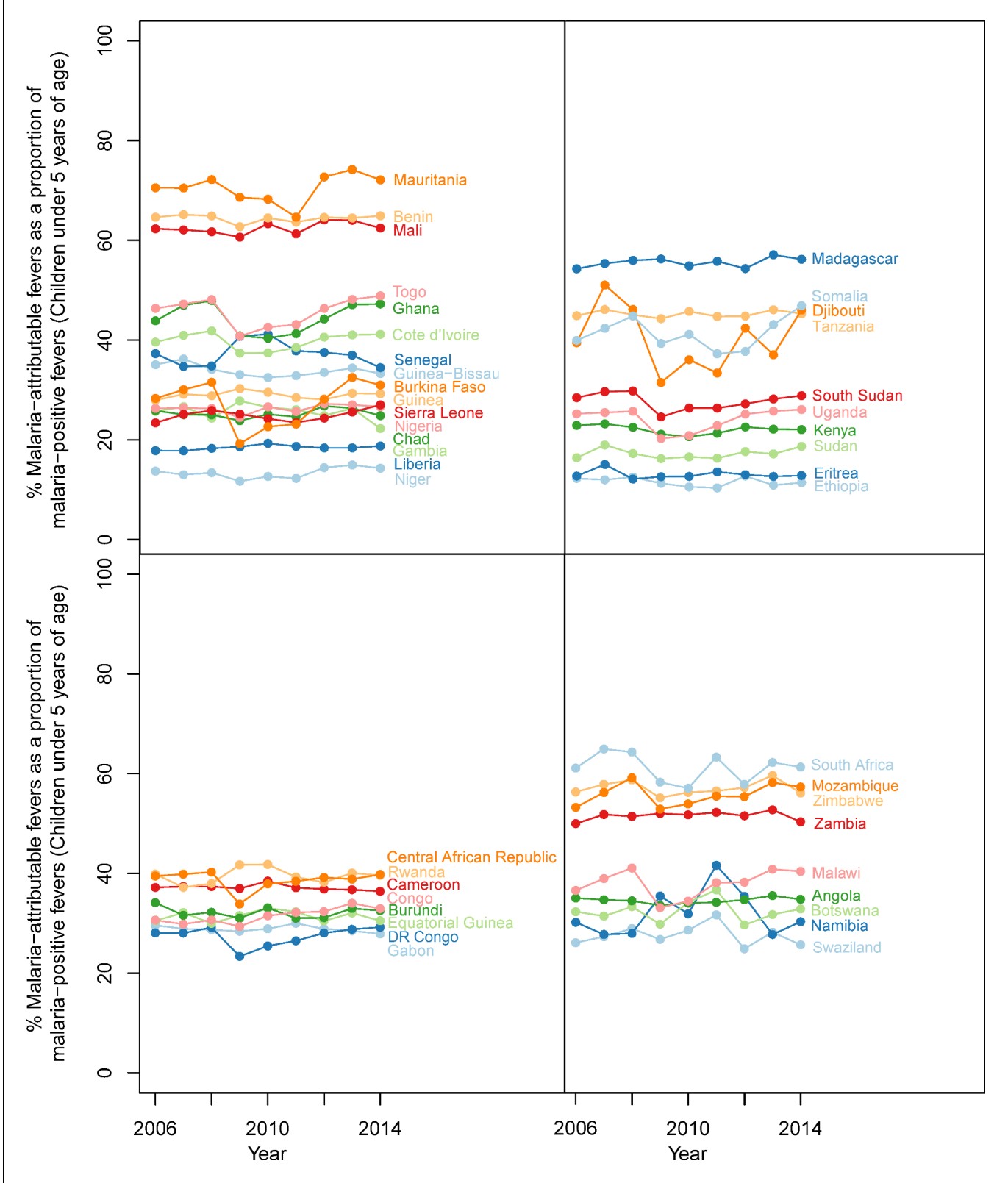

**Figure 3.** Malaria-attributable fevers as a proportion of malaria-positive fevers (children < 5 years of age, 2014). Plotted values are the population-weighted mean for 43 sub-Saharan African countries over the study period, 2006–2014. Countries have been grouped by region to improve clarity.
DOI: https://doi.org/10.7554/eLife.29198.005

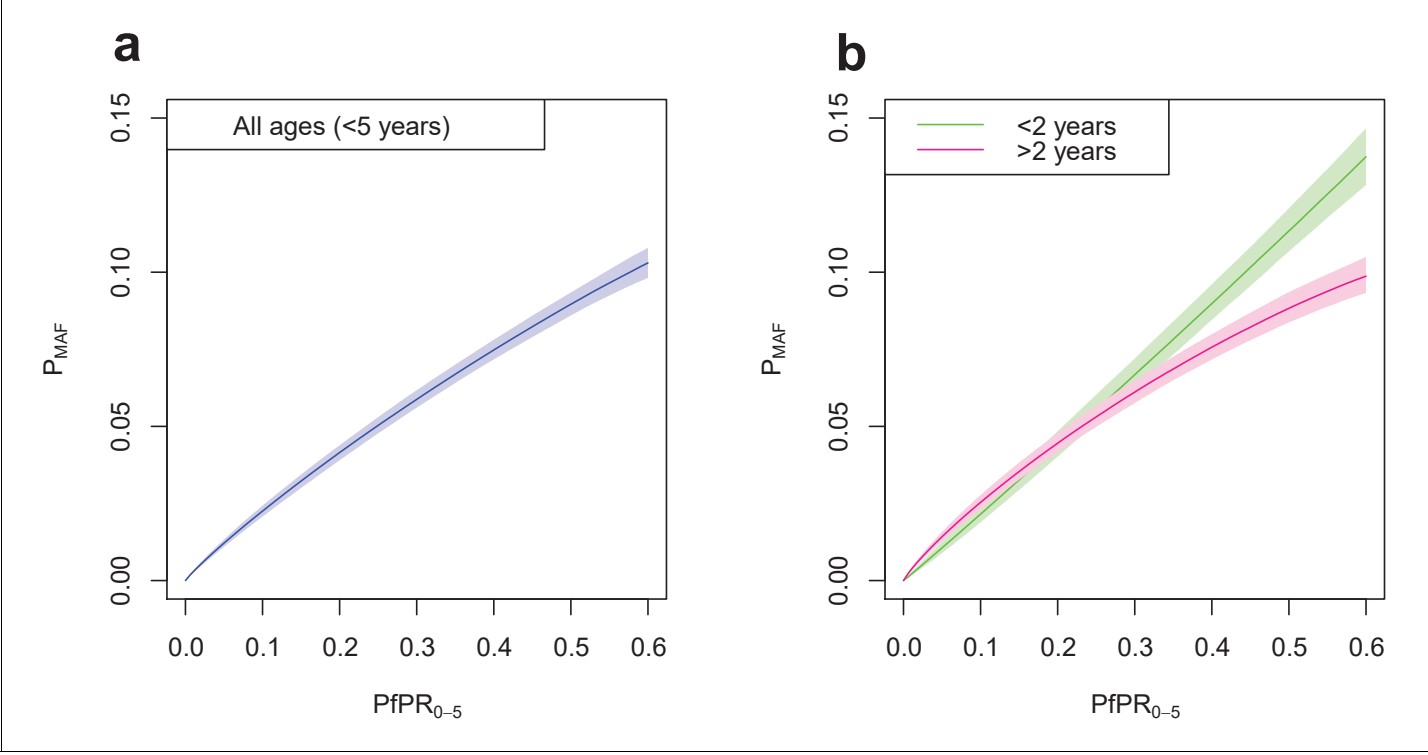

**Figure 4.** Final fitted relationship between $PfPR_{0-5}$ and probability of a malaria-attributable fever (MAF) in the past two weeks. (a) shows this relationship in children under five years of age, and (b) disaggregated into children under 2 years of age, and children aged 2–4 years. The probability of MAF in the past two weeks is greater for children under 2 years of age than for children above 2 years of age in areas with a $PfPR_{0-5}$ higher than approximately 0.3. Median values of the posterior distribution are shown, with shaded 95% credible intervals.

DOI: https://doi.org/10.7554/eLife.29198.006

with overlaid estimations of clinical incidence over the duration of the past zero to two, and two to four weeks, with increasing $PfPR_{0-5}$ from an ensemble of transmission models (*Cameron et al., 2015*). The breakdown of the contribution of NMFI, malaria-coincident fevers, and fevers without a patent malaria infection to all-cause fever varied considerably by country; this breakdown is displayed in detail for 2014 in *Supplementary file 5*.

## Discussion

The analysis presented here shows that although the proportion of fever cases that are accompanied by an RDT-patent *P. falciparum* infection remains high, only approximately a third of these fevers are causally attributable to malaria. The majority of febrile illness in Africa is caused by pathogens other than *P. falciparum* malaria, even in areas where malaria is highly endemic, and the proportion of all fevers caused by NMFI has risen since 2006. We estimate here that over a typical two-week period in 2014 one in four children under five years old residing in the limits of stable malaria transmission will suffer a fever not caused by *P. falciparum* malaria, whereas only one in every 32 children will suffer a fever directly caused by *P. falciparum*. We show that current estimations of malaria burden in Africa based on RDT-positive cases of fever may be overestimating the burden by up to two-thirds, and the level of overestimation is likely to be highly heterogeneous between different countries (*Figure 3*). For example, countries such as Niger, Gabon and Nigeria, where fewer than 21% of malaria-positive fever cases are causally attributable to malaria, may be substantially overestimating the morbidity caused by malaria (and systematically underestimating the burden of other diseases that may be co-infecting and causal). Additionally, these results have implications for the effectiveness of treatment received at point of care, as a substantial proportion of RDT-positive fevers are likely to be coincident with and caused by a NMFI. Our estimations of MAF prevalence in the past two weeks are comparable to estimations of clinical incidence from an ensemble of transmission models

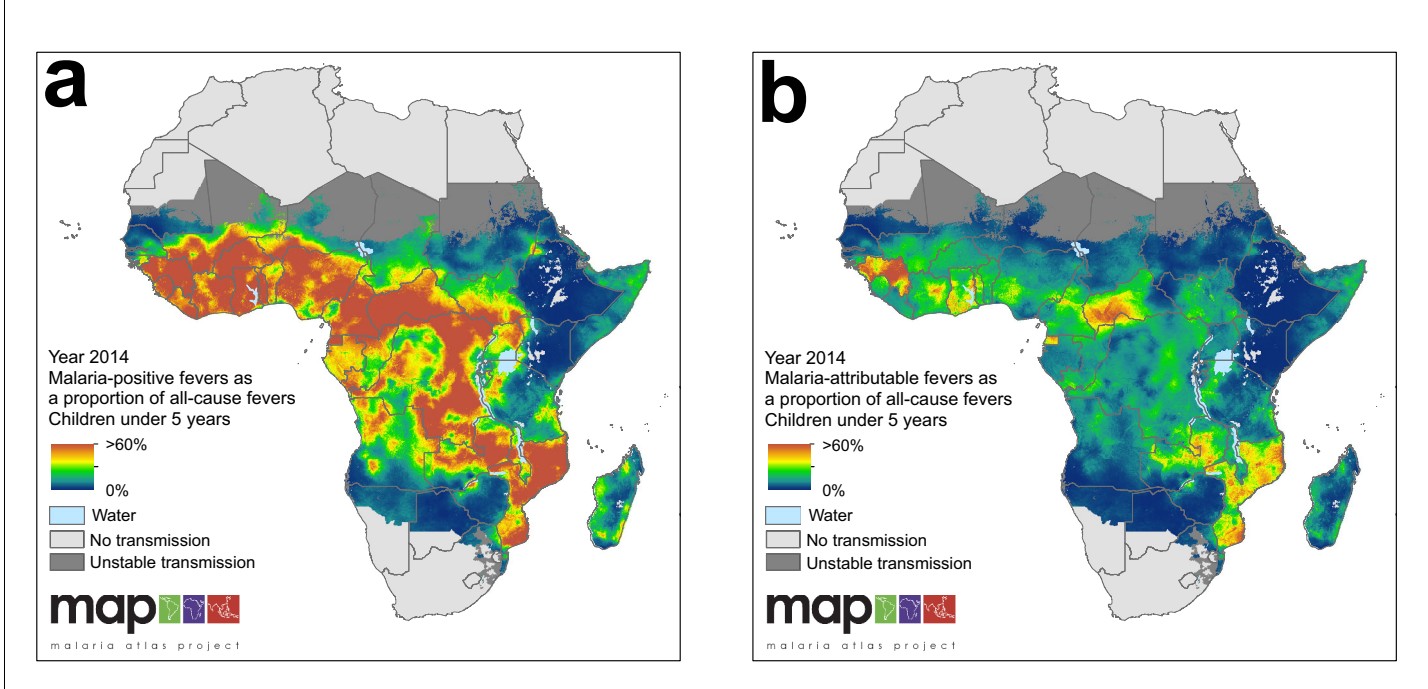

**Figure 5.** (a) Predicted malaria-positive fevers as a proportion of all fevers; (b) predicted malaria attributable fevers (MAF) as a proportion of all fevers. Both maps are shown for the year 2014, for children under 5 years of age and bounded by the limits of stable *P. falciparum* transmission.
DOI: https://doi.org/10.7554/eLife.29198.007

The following figure supplement is available for figure 5:

**Figure supplement 1.**
DOI: https://doi.org/10.7554/eLife.29198.008

(*Figure 5—figure supplement 1*) (*Cameron et al., 2015*) although our estimates match more closely to transmission model estimations of the prevalence of febrile illness lasting the duration of the past two-to-four weeks, rather than the past zero-to-two. For 13 of the 38 household surveys used in this analysis, the number of days since the onset of the child's fever was available. Children in these surveys had a fever for on average 5.38 days (±3.85 days) prior to the interview. This disparity between fever duration in our estimates and transmission model estimates can perhaps be explained by the cross-sectional nature of the surveys; the transmission models estimate the full course of the fever without treatment, whereas the household surveys aim to curtail the fever at the interview by administering appropriate medication to febrile children.

The total number of fever cases accompanied by an RDT-patent *P. falciparum* infection declined substantially over the study period (from 48.5% in 2006 to 35.7% in 2014), linked with declining $PfPR_{0-5}$ over the same time (*Bhatt et al., 2015*; *Smith et al., 2007*). The only country with a noteworthy increase in malaria-positive fevers over the study period was Gabon, which also experienced an increase in $PfPR_{0-5}$ between 2006 and 2014.

This modelling approach makes use of household survey datasets in a novel way; no previous attempts have been made to measure childhood fever prevalence (and the malaria/non-malaria causality of the fever) from surveys recording two-week fever history. This approach has some drawbacks: it relies on two-week recall of fever from the child's caregiver, the accuracy of which can be subject to biases such as (i) length of time since the fever; (ii) the child's frequency of febrile episodes, (iii) the economic status of the family, and (iv) even the child's sex (*Das et al., 2012*; *Rockers and McConnell, 2017*). We also include fever caused by *P. vivax* malaria as a NMFI by definition, as not all household surveys tested for *P. vivax* with RDT and thus only *P. falciparum* outcomes were used in this analysis. This is unlikely to have a major effect on most of Africa due to high Duffy negativity resulting in unstable *P. vivax* transmission, and very low levels of stable transmission in Madagascar and the horn of Africa (*Gething et al., 2012*; *Howes et al., 2011*). One drawback of

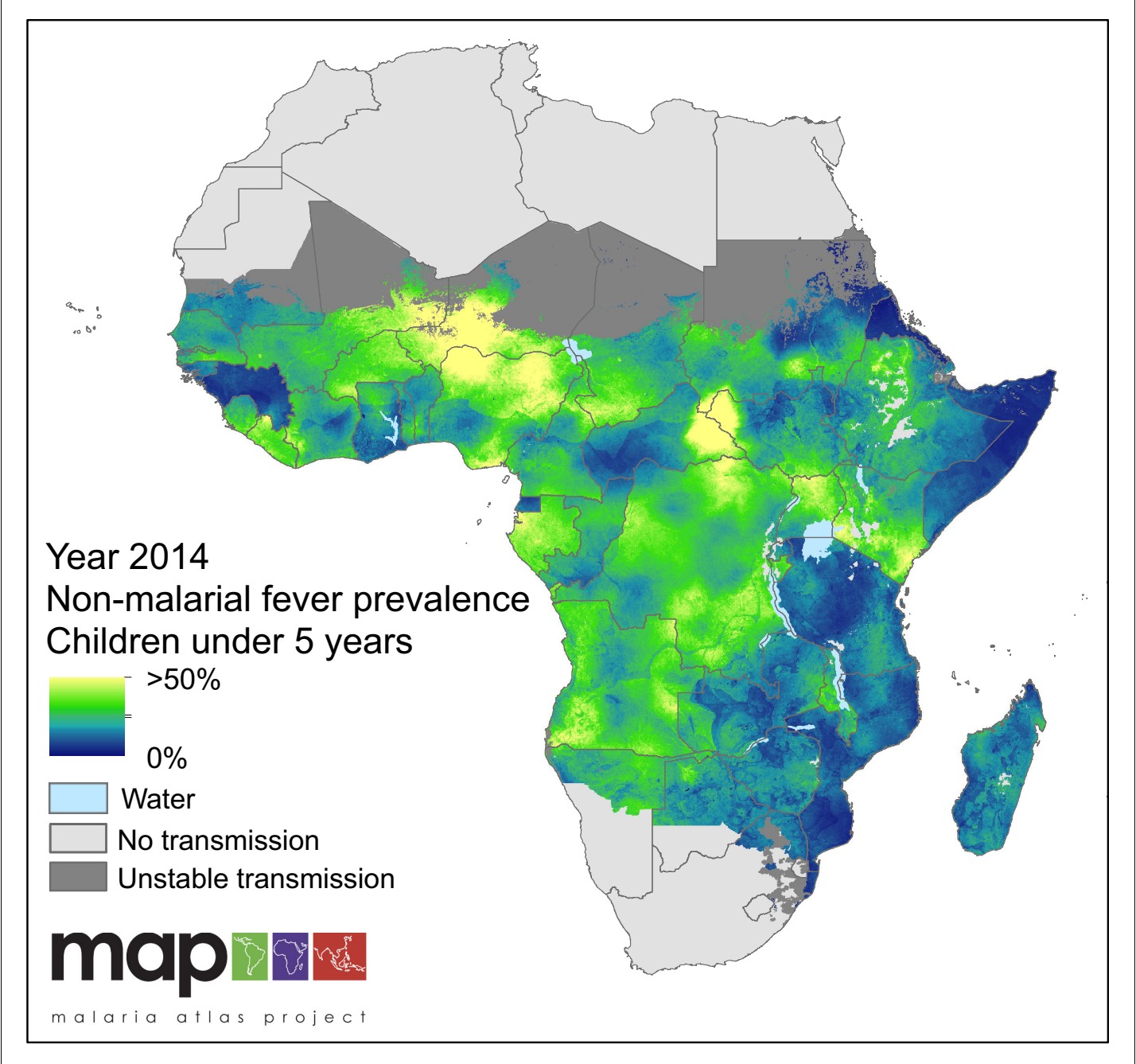

**Figure 6.** Predicted non-malarial febrile illness (NMFI) prevalence in children under 5 years of age. NMFI prevalence is defined as the sum of the prevalence of febrile illness without a *P. falciparum* malaria infection and the prevalence of febrile illness coincident with, but not caused by, a P. falciparum malaria infection (MCF), for children under 5 years of age and bounded within the spatial limits of stable *P. falciparum* transmission in 2014.
DOI: https://doi.org/10.7554/eLife.29198.009

using household survey data to generate predictions such as those presented here is that the survey data interviews children on one day within the time frame that the survey is conducted (typically two to four months). It is not necessarily the case that children who present a patent malaria infection but no history of infection over two weeks prior to their survey date have never, or will never, develop a malaria-attributable fever, and may have already done so outside of the two-week recall period of the survey interview. Additionally, co-infections may lead to a fever that would not have occurred had the child only had a monoinfection with malaria or the NMFI (*French et al., 2001*;

*Hartgers and Yazdanbakhsh, 2006*).This study does not consider the effects of co-morbidity in this sense, and fevers that fall into this category are accounted for within NMFI. We also incorporate no predictor variables at resolutions below the pixel-level (e.g. individual or household level), which are likely to be significantly predictive of both MAF and NMFI prevalence. Predictors such as household wealth and other associated health indicators are available within the household survey datasets, but owing to the heterogeneity of individual level variables measured in different surveys and the lack of a common set of information directly representative of those variables outside of surveyed countries, we have not incorporated them in the current model.

The low prevalence of malaria-positive fevers at all levels of *P. falciparum* endemicity suggests that passive case detection alone may not be enough to halt transmission of malaria, as many infected children are afebrile at any one time (regardless of whether their fever is attributable to the malaria infection or not). Further, this study estimates MAF prevalence in children less than two years of age, and between two and four years of age. We find that the relationship between MAF prevalence and *Pf*PR is not equivalent across all age groups (*Figure 4*), with children under two years of age having a higher probability of developing a malaria-attributable fever than children older than two in areas where *Pf*PR$_{0-5}$ is greater than 30%. For older children living in high *Pf*PR locations, a higher proportion of the population will have acquired immunity to malaria infection (*Doolan et al., 2009*), and therefore may be less likely to present with a malaria-attributable fever. Although there is uncertainty over how long malaria immunity lasts (*Filipe et al., 2007*; *Wipasa et al., 2010*), in areas of rapid decline of malaria prevalence that are now moving towards elimination, this residual acquired immunity may present problems for elimination if programs rely on passive case detection and treatment to halt transmission. Again, the limitation of our estimates which use a two-week window for reporting of symptoms means that, in order to assert with confidence that passive case detection would be insufficient to halt transmission, evidence for a high prevalence of never-symptomatic infections would also be necessary.

Despite all malaria-endemic countries now adopting the official policy of parasite-based diagnosis of suspected malaria cases before issuing antimalarials, a large number of suspected malaria cases still do not receive a diagnostic test before treatment (*Hertz et al., 2013*; *World Health Organization, 2015*). This proportion has decreased over the time period of this study, from 64% of suspected cases not receiving a diagnostic test in 2005 to only 35% in 2014 (*World Health Organization, 2015*). The reasons for this high proportion of suspected cases not receiving parasite-based diagnosis lies predominantly in resource limitation in rural health clinics in malaria endemic countries (*Ghai et al., 2016*). This remaining proportion of NMFI that is assumed to be caused by malaria, and thus treated with an antimalarial, exacerbates the problem of over-prescription of antimalarials. By enumerating the proportion of NMFI, this study provides evidence of the on-going need to test febrile individuals for parasitaemia, but can also be used identify different geographic and malaria-burden settings where administering antimalarials in the absence of a positive test for *P. falciparum* is least warranted.

This suite of maps demonstrates the need for better diagnostic tests for other pathogens causing NMFI in Africa, which will in turn lead to a better understanding of the range of diseases that make up the non-malarial fraction of all-cause fever prevalence. Whilst the spatial distribution of pathogens that cause NMFI is largely unknown, occurrence data for these diseases are becoming more widely available, particularly in the context of identifying diseases with similar symptomatic presentations as malaria. The Worldwide Antimalarial Resistance Network (WWARN) now collects clinical evidence of non-malarial febrile illness (NMFI) caused by neglected tropical diseases from peer-reviewed case studies of fever diagnoses from across the malaria-endemic world (*Worldwide Antimalarial Resistance Network, 2017*). The collation of these studies is designed to give an overview of the presence of fever-causing pathogens in areas of overlapping endemicity with malaria, but cannot presently be used to quantify the spatiotemporal distribution of NMFI, nor can it be used to measure the individual contribution of each disease as an underlying cause of febrile illness. For example, a major review of 146 case studies in the Mekong region of Southeast Asia (*Acestor et al., 2012*), showed that misdiagnosis and mistreatment of the fever was a common outcome for individuals who were co-infected with malaria and other diseases with overlapping symptoms (e.g., Dengue fever virus, *Orientia tsutsugamushi* and *Rickettsia* species). In addition to implicating potential overestimation of the burden of malaria in the past, the results of this study suggest that current WHO guidelines for integrated management of childhood illness (IMCI) is

currently suboptimal for the treatment of non-malarial fever. Whilst treating positive malaria infections is the correct strategy for slowing malaria transmission, removing the malaria-attributable fevers from the population, and reducing the reservoir of infected individuals that perpetuate transmission; current procedures may lead to systematic mismanagement of NMFI. This study shows the need for increased efforts to develop and distribute routine diagnostics for NMFI-causing pathogens.

## Materials and methods

### Household survey data

Thirty-eight cross sectional, nationally-representative georeferenced surveys of malaria prevalence in children less than five years of age across 24 countries in sub-Saharan Africa were obtained from the Malaria Atlas Project database (*Guerra et al., 2007*; *Malaria Atlas Project , 2017*). These surveys originate from a variety of sources such as the DHS Program, UNICEF Multiple Indicator Cluster Surveys, and national Ministries of Health in malaria-endemic countries. Full details of the surveys used can be found in *Supplementary file 2*.

In these surveys, a blood sample was taken from any children present and tested for malaria parasites with both a RDT and Giemsa-stained microscopy. Here, we used the RDT-derived diagnostic outcome as an indicator of malaria positivity within the last two weeks, to lessen the possibility of a recent infection having been cleared via artemisinin combination therapy (ACT) (and therefore presenting as a negative result by microscopy). RDTs offer a more accurate representation of two-week infection than microscopy due to the persistence of histidine-rich protein II in the blood for up to 14 days after parasite clearance with ACT (*Mayxay et al., 2001*). Diagnostic and fever history outcomes from a total of 155,369 children that were aggregated within $5 \times 5$ km raster cells, resulting in a total 10,606 data points. In addition to the diagnostic and fever history outcomes, the wealth quintile (i.e. poorest, poorer, middle, richer, richest) of the individual's household was also extracted for use as an household-level predictor of fever in the final multinomial model.

### Covariates

Independent model-based spatiotemporal predictions of both the prevalence of *P. falciparum* infection ($Pf$PR$_{0-5}$) and for the environmental suitability for background (non-malarial) fevers in children under five years of age across Africa for each year within the span of the household survey data (2006–2015) were constructed as covariates for use in the final multinomial model.

The $Pf$PR$_{0-5}$ covariate was generated for each year using the $Pf$PR$_{2-10}$ modelled predictions of slide-patent African $Pf$PR from the Malaria Atlas Project (*Bhatt et al., 2015*), age-standardised to 0–5 years using the 'agestand' package in R, based on a previously described age-standardisation procedure for $Pf$PR (*Smith et al., 2007*), and transformed to predict RDT-based prevalence via an microscopy-to-RDT relationship identified in earlier work (*Mappin et al., 2015*).

Prior to this study, no mapped estimates of environmental suitability for background fever existed for the African continent, hence a modelled estimate of environmental suitability for background fever was constructed using a boosted regression tree (BRT) model. The BRT approach generates a flexible regression model in two steps. Initially, response data are split recursively by predictor variables, where predictor variables that maximise the partitioned response data's homogeneity are selected more frequently in subsequent steps. This is followed by a 'boosting' step, where the regression tree models are combined to reduce the model's overall predictive deviance (*Elith et al., 2008*), and at each iterative step cross-validated against a randomly held-out subset to avoid overfitting (*Bhatt et al., 2013*).

Firstly, a training dataset was constructed, consisting of a subset of the household survey dataset where survey cells with ten or fewer tested individuals were removed in order to increase the reliability of observed prevalence values. This procedure reduced the final number of data points in the training dataset to 5320. Background fever prevalence was obtained at each data point through aggregation of individuals with a fever in the previous two weeks without a patent malaria infection.

A large suite of potential environmental and socio-demographic covariates was assembled, as described elsewhere (*Weiss et al., 2015*). Briefly, $5 \times 5$ km resolution surfaces across the African continent of different environmental and sociodemographic covariate types—elevation, land cover,

population density, precipitation, enhanced vegetation index (EVI), land surface temperature (LST), *P. falciparum* temperature suitability index (TSI) (*Weiss et al., 2014*), tasselled cap wetness (TCW), tasselled cap brightness (TCB), nighttime lights, Global Urban Footprint (*Esch et al., 2012*), and national-level World Bank indicators of poverty and healthcare quality (*World Bank, 2017*)—were obtained from the Malaria Atlas Project database and leveraged by constructing pixel-level spatial summaries (maximum, mean, minimum, standard deviation and range). These summarisations produced a total of 986 spatially- and temporally-contemporaneous covariates from which we extracted results for each of the 5320 georeferenced response data points.

To reduce the number of covariates an exploratory BRT model was constructed using background fever prevalence as the response variable and the covariates as predictor variables using the 'gbm. step' function in the 'dismo' package in R (*Elith et al., 2008*). The BRT model was parameterised with a tree complexity of 5, a learning rate of 0·05, a hold-out fraction of 0·05 and an error structure within the family 'gaussian' after a suitable inverse-sigmoid transform of the observed background fever prevalences. Each of the 986 covariates was ranked by their contributions to the fitted exploratory model. To reduce redundancy, each variable within the assembled suite of covariates was tested for collinearity with every other variable. Covariates were considered collinear if a correlation of greater than 0.7 existed between the two covariates, and when collinearity was identified only higher-ranked covariate was retained. This approach left a final dataset of 174 predictor variables for each of the 5320 response data points.

The final model, using the reduced set of 174 non-collinear predictor variables, was fitted with 2000 trees, a tree complexity of 5, and a reduced learning rate of 0.01 (reducing the learning rate improves model performance [*Elith et al., 2008*]). A list of the 174 predictor variables and each variable's contribution to the final model used can be found in *Supplementary file 3*. Predictions for the environmental suitability for background fever amongst children under five years old for each month between January 2006 and December 2014 across the area of stable malaria transmission in Africa were generated using the fitted BRT model and the 174 predictor variables. For a number of predictor variables, data were unavailable in November and December 2014, so data from the same month in the previous year were used instead. Yearly predictions for 2006–2014 were generated as a mean of each monthly prediction for use in the multinomial model. Gaps in the resulting predicted layers, caused by small gaps in the remotely-sensed covariate layers, were filled by recursively scanning through each raster cell by cell, and filling in cells with no data with the mean of the first layer of surrounding cells until no gaps remained.

## Multinomial model

A training dataset for the multinomial model was constructed from all 10,606 points in the household survey dataset. Of these points, 25% (2,652) were selected for the holdout dataset, where probability of selection was directly proportional to the distance of the nearest neighbouring point, to avoid clustering of hold-out points in areas of dense data coverage. The remaining 75% of points (7,954) were used in the training dataset.

The observed data at each site, $i$ (a 5 × 5 km pixel), with $N_{\text{tot.}}^i$ surveyed individuals may be represented as a two-by-two categorical table of counts according to fever status (febrile or afebrile) and RDT-based *P. falciparum* parasite status (positive or negative). The hierarchical Bayesian model we construct here thus takes a top-level likelihood with multinomial distribution having four unknown parameters, $\{q_1^i, q_2^i, q_3^i, q_4^i\}$, that describe the expected proportion of counts in each category; i.e.,

$$\left[N_{\text{feb. \& pos.}}^i, N_{\text{feb. \& neg.}}^i, N_{\text{afeb. \& pos.}}^i, N_{\text{afeb. \& neg.}}^i\right] \sim \text{Multinomial}\left[\{q_1^i, q_2^i, q_3^i, q_4^i\} | N_{\text{tot.}}^i\right].$$

Since each expected proportion must lie between zero and one while their joint sum is strictly unity, the effective dimension of unknown parameters here is three, with constraint to the standard 3-simplex. Our parameterisation takes three components accordingly, which we define so as both to respect these constraints and to represent the key targets of our inference as shown in *Table 1*.

Here $p_{\text{pos.}}^i$ is the expected prevalence of malaria parasites in children under five years of age, $p_{\text{bg}}^i$ is the expected prevalence of background fever (within a two-week window) in the same cohort, and $r_{\text{maf}}^i$ gives the expected proportion of *P. falciparum* positives reporting a fever attributable to malaria. By convention (and in line with our objectives) we define the prevalence of these 'malaria-

**Table 1.** Four-way infection outcome table and formulae for deriving targets of inference.

| | Pf pos. | Pf neg. |
|---|---|---|
| Febrile | $q_1^i = p_{\text{pos.}}^i \left( r_{\text{maf}}^i \left( 1 - p_{\text{bg}}^i \right) + p_{\text{bg}}^i \right)$ | $q_2^i = \left( 1 - p_{\text{pos.}}^i \right) p_{\text{bg}}^i$ |
| Afebrile | $q_3^i = p_{\text{pos.}}^i \left( 1 - r_{\text{maf}}^i \right) \left( 1 - p_{\text{bg}}^i \right)$ | $q_4^i = \left( 1 - p_{\text{pos.}}^i \right) \left( 1 - p_{\text{bg}}^i \right)$ |

DOI: https://doi.org/10.7554/eLife.29198.010

attributable fevers', $p_{\text{maf}}^i$, as $p_{\text{maf}}^i = p_{\text{pos.}}^i . r_{\text{maf}}^i \left( 1 - p_{\text{bg}}^i \right)$ ; that is, excluding causal fevers 'coexisting' with a background fever.

Our representation of the local parasite prevalence in children under five years of age takes the form of a spatial generalized linear model (GLM) in which the logit transform of predicted prevalence from the MAP $Pf\text{PR}_{2\text{-}10}$ space-time cube (**Bhatt et al., 2015**), age-standardised to $Pf\text{PR}_{0\text{-}5}$ (**Smith et al., 2007**), is used as a linear predictor for the logit transform of the under-five year old prevalence, augmented with a latent (spatial) Gaussian random field (GRF). That is,

$$\text{logit } p_{\text{pos.}}^i = \text{logit } Pf\text{PR}_{\text{MAP}}^i + f_{x(i),}$$

$$f \sim GRF_{\theta.}$$

An identical model is used for the rate of background fevers in this cohort, except that the predictor variable is now the logit of the environmental suitability for background fevers predicted by the BRT model outlined above. Hence,

$$\text{logit } p_{\text{bg}}^i = \text{logit } \text{FEV}_{\text{BRT}}^i + g_{x(i),}$$

$$g \sim GRF_{\phi.}$$

Our model for the proportion of parasite positives with a causal malaria fever is a simple quadratic dependence on the local prevalence, motivated by the form observed in a previously described transmission model (**Maire et al., 2006**; **Ross et al., 2006**; **Smith et al., 2006a**; **Smith et al., 2006b**). Namely,

$$\text{logit } r_{\text{maf}}^i = \delta + \psi \times \text{logit } p_{\text{pos.}}^i + \xi \times \left( \text{logit } p_{\text{pos.}}^i \right)^2.$$

For computational tractability we adopt a Gaussian Markov Random Field (GMRF) approximation to the continuous GRF components of our model using the version for Matern GRFs identified by Lindgren et al. (**Lindgren et al., 2011**) from the INLA package (**Rue et al., 2009**). Implementation of this model requires construction of a mesh-based tessellation enclosing the African continent (plus Madagascar) which we also perform with the INLA package using 6280 mesh nodes. We fixed the smoothness of both GMRFs used in our model to $\nu = 1$ ($\alpha = 2, d = 2$) and set (weakly informative) standard Normal priors on the logarithm of the hyperparameters for each, $\kappa$ (i.e., $2\sqrt{2} \times \text{inverse range}$) and $\tau$ (i.e., inverse variance). Finally, we complete our model by placing priors on the intercept and two slope coefficients of the $\text{logit } r_{\text{maf}}^i$ model:

$$\delta \sim \text{Normal}\left[-2, 1.0^2\right], \quad \psi \sim \text{Normal}\left[0, 0.25^2\right], \quad \text{and} \quad \xi \sim \text{Normal}\left[0, 0.25^2\right].$$

Model fitting was performed with the TMB package (**Kristensen et al., 2015**) for automatic differentiation, which returns functions for the likelihood plus gradient and Hessian matrices with respect to all parameters of our model after (approximate) marginalisation over the random fields via a Laplace approximation. These are then plugged into the 'nlminb' function in R and optimised to find the empirical Bayes solution with the local Hessian used to form approximate credible intervals as summarized in **Supplementary file 4**. Additional model fits to response data disaggregated by age

(children over and under two years of age) were conducted to assess the age-effect on the likelihood of developing MAF with varying *Pf*PR.

Model validation was perform via an initial fit of the above model to a sub-sample (75%) of the full dataset using a spatial leave-one-out cross validation procedure (*Le Rest et al., 2014*). The probability integral transform diagnostic (*Angus, 1994*; *Diebold et al., 1997*) for multinomial probabilities at the holdout sites estimated from the empirical quantiles of draws from the posterior predictive was used as a qualitative check on the calibration of our posterior uncertainties (*Figure 1—figure supplement 1*). All code for the multinomial model and the BRT model are available on GitHub (*Dalrymple, 2017*; a copy is available at https://github.com/elifesciences-publications/MAF-NMFI).

## Additional information

### Funding

| Funder | Grant reference number | Author |
|---|---|---|
| Medical Research Council | Doctoral Training Grant | Ursula Dalrymple |
| Bill and Melinda Gates Foundation | H5R00640 | Ewan Cameron Samir Bhatt Daniel J Weiss Peter W Gething |
| Bill and Melinda Gates Foundation | H5R00690 | Ewan Cameron Samir Bhatt Daniel J Weiss Peter W Gething |

The funders had no role in study design, data collection and interpretation, or the decision to submit the work for publication.

### Author contributions

Ursula Dalrymple, Conceptualization, Resources, Data curation, Formal analysis, Funding acquisition, Validation, Investigation, Visualization, Methodology, Writing—original draft, Project administration, Writing—review and editing; Ewan Cameron, Conceptualization, Resources, Software, Formal analysis, Supervision, Validation, Investigation, Methodology, Writing—original draft, Writing—review and editing; Samir Bhatt, Methodology, Writing—review and editing; Daniel J Weiss, Data curation, Methodology, Writing—review and editing; Sunetra Gupta, Supervision, Writing—review and editing; Peter W Gething, Conceptualization, Resources, Supervision, Methodology, Writing—original draft, Writing—review and editing

### Author ORCIDs

Ursula Dalrymple (iD) http://orcid.org/0000-0001-6206-3777

### Decision letter and Author response

Decision letter https://doi.org/10.7554/eLife.29198.019
Author response https://doi.org/10.7554/eLife.29198.020

## Additional files

### Supplementary files

• Supplementary file 1. Yearly population-weighted percentages for all-cause fever prevalence, and malaria-attributable fever, malaria-positive fever, and non-malarial febrile illness within all-cause fever amongst population at risk in children under 5 years of age in malaria-endemic Africa, 2006–2014.
DOI: https://doi.org/10.7554/eLife.29198.011

• Supplementary file 2. Household survey data used in analysis.
DOI: https://doi.org/10.7554/eLife.29198.012

• Supplementary file 3. Covariates used in BRT, listed by contribution to the final model.
DOI: https://doi.org/10.7554/eLife.29198.013

• Supplementary file 4. Model parameters and credible intervals of the final multinomial model; final model coefficients.
DOI: https://doi.org/10.7554/eLife.29198.014

• Supplementary file 5. National breakdown of contribution of NMFI, MAF and MCF (malaria-coincident fever) to all-cause fever in 2014. NMFI is represented here as a fever without a patent malaria infection, MCF as a fever with a patent malaria infection where the fever is caused by a co-infection with an NMFI, and MAF where the malaria infection is the sole cause of the individual's fever. This file is supplementary to *Figure 5*.
DOI: https://doi.org/10.7554/eLife.29198.015

• Transparent reporting form
DOI: https://doi.org/10.7554/eLife.29198.016

## Major datasets

The following previously published datasets were used:

| Author(s) | Year | Dataset title | Dataset URL | Database, license, and accessibility information |
|---|---|---|---|---|
| UNICEF MICS | 2011 | UNICEF MICS Ghana 2011 | https://www.unicef.org/ghana/Ghana_MICS_Final.pdf | Available upon request from UNICEF MICS (http://mics.unicef.org/surveys) |

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
