## [Decision Letter]

Thank you for submitting your article "Quantifying the contribution of malaria versus other causes to febrile illness amongst African children" for consideration by *eLife*. Your article has been favorably evaluated by Prabhat Jha (Senior Editor) and three reviewers, one of whom, Mark Jit, is a member of our Board of Reviewing Editors. The following individual involved in review of your submission has agreed to reveal their identity: Thomas Eisele.

The reviewers have discussed the reviews with one another and the Reviewing Editor has drafted this decision to help you prepare a revised submission. We agree that while the manuscript is not acceptable in its current state, we believe that it has potential and would like to see a version with major revisions.

In particular, the reviewers and I agreed that you took an interesting approach to an important public health question, using a large, rich dataset. However we felt that there were methodological shortcomings in the model (or at least in the current description of it) that needed to be addressed. The main issues that need to be addressed are listed below.

1) Model validation

Although it is mentioned that the fever model has been validated (subsection “Model overview”), it is not clear how this was actually done. Part of the problem is that you have not presented any of the original data on fevers and malaria prevalence. This makes it impossible to tell the goodness of fit.

Given that *eLife* has no space or figure restrictions, the following data plots would ideally be shown:

- The raw 2x2 table showing all fevers versus RDT positivity (data and predictions), which is really interesting and is key to the message of the paper. This should also be plotted by country and transmission intensity.

- Model-predicted vs. observed all cause fever by pixel or by country. A fairly small subset of the available DHS and MICS surveys are used, presumably because the rest do not have any data on RDT positivity (and/or GPS?). However these other surveys do contain data on all-cause fever and therefore would be an excellent source of validation data for this aspect of the model.

- A plot of the final fitted relationship between fever prevalence, parasite prevalence and the malaria-attributable fraction (and data on the 1st two variables)

- Comparison with observed vs. predicted fever prevalence. These data are not presented, and one of the reviewers who checked could not reconcile some of their predictions with the available data online. E.g. Results subsection “Prevalence of all-cause fever”: fever prevalence in Liberia is quoted as 51.7% in 2014, but on the DHS survey website only a Liberia survey in 2013, which gave fever prevalence as 28.6%. Similarly the most recent data from Niger (DHS 2012) shows a fever prevalence of 14.2%, but the quoted value by the authors is 57.2%. Please could you check and explain the reasons for any discrepancies.

- Related to those countries with the highest and lowest prevalence of fever (subsection “Prevalence of all-cause fever”): predicted values are given for Eritrea, Niger, Botswana and Zimbabwe. But in the supplementary data table (Supplementary file 2), the list of surveys does not include any data from any of these countries. Is this correct, or is the data list incomplete? If correct it seems concerning that the least certain estimates produce the outlying predictions. This should warrant some reinvestigation of the model fit. We suggest at least comparing the predictions against available fever prevalence estimates, even if not coupled with RDT data (as mentioned above).

More generally, external validation should be performed by comparing model results to datasets not used to parameterise it. For instance you mention that case-control studies, transmission models etc. have been used to estimate the attributable fraction of malaria in fever – it would be useful to compare the model results with these studies.

For transparency and to comply with *eLife* policy for mathematical models, you should provide the initial and final set of model equations (including coefficients) and the code used to select the final model, either in Supplementary Materials or in a suitable online repository such as github. Major data sets used that are not already in the public domain should also be provided unless there are compelling scientific or ethical reasons not to.

2) Extending the model beyond environmental variables

Although in the fifth paragraph of the subsection “Covariates” it mentions that socio-demographic predictors of fever were considered, all the examples in the paragraph and in Supplementary file 3 are environmental. This seems like quite a large omission, especially given that fever is self-reported. As well as many varied causes of fever, cultural perceptions of fever are important here (and language, e.g. some languages do not have a specific word for fever). We were not clear how well environmental variables could be expected to predict this, especially given that you are extending fever predictions into countries for which there are no data.

Besides cultural factors, a list of suitable covariates may include socioeconomic factors such as nutrition, crowding, mother's education, access to clean water etc. Many of these are available in DHS/MICS data but it is difficult to see how they were used. Indeed, given that the final model contained 167 predictor variables, it is difficult to see why there was any human selection at all. The authors have not tried to justify any of their chosen variables based on causality arguments, so would it not make more sense to use the entire DHS dataset as predictors, and then let variable selection algorithms winnow this down?

We are also concerned about the apparent lack of variables at the household or even individual level. This may be a problem if being malaria-positive is (positively or negatively) associated with having a fever beyond what can be explained by the spatial covariates examined. For instance, within a particular town or village (with homogeneous environmental variables) there will be poorer households who are more likely to be both malaria-positive and to have non-malaria fevers. Within the household there will be further associations in distribution due to the age, gender, birth order, genetic makeup etc. of individuals. Perhaps this has been taken into account, but it is not clear how this happened.

In general, we get the impression that the ecology of the pathogen, its environment and insect host is well-described, but the epidemiological, immunological, cultural and socioeconomic determinants of disease within the human host are either less well captured or at least less well explained. Perhaps you should include someone with clinical or at least public health training in your authorship list.

3) Scope

There are a number of areas where we believe that the scope and implications of the results may be overstated.

- You need to clarify that this study aims at improving burden estimates of uncomplicated malaria, and not case management policy. WHO clearly recommends, as do all national policies in African countries with endemic malaria, that all fevers/suspected malaria presenting at facilitates in malaria endemic areas should receive a laboratory diagnosis for malaria, and if positive treated with the first-line antimalarial. This doesn't mean the attending health professional cannot go on to treat other presenting illnesses and symptoms. But even if the fever is not directly attributable to the Pf infection at that time, it should be treated. This needs to be made clear in the paper. You should stick to how these findings impact the overall epidemiology of fever illness among children in Africa, and not make recommendations or draw conclusions from this study in the discussion for malaria case management (or IMCI) policy.

- You argue that the results of their work will improve burden estimates. We find this to be somewhat of a 'straw man' attack, as to our knowledge no burden estimates have been based on an RDT positive child with a history of fever in the past 2 weeks. Neither WHO GMP, MAP nor GBD uses such a method.

- You present cross-sectional household survey data that measures a 2-week (or there about) RDT period prevalence based on persisting HRP2 antigenemia from a Pf infection, plus an overlapping fever history based on the recall by the mother/caregiver. Their primary results suggest a large proportion of these fevers are not directly attributed to the underlying Pf infection. While this seems an appropriate interpretation of the results and in line with malaria epidemiology, the cross-sectional nature of the study is a major limitation. You need to make note that the underlying Pf infection likely would have resulted in at least 1 parasite-attributable fever, likely in the first month of the infection, and additional parasite-related fevers will likely occur, especially if a new infection occurs on top of the existing infection (just based on the malaria therapy data). So the timing of the observed RDT+ and fever is important in understanding the true relationship between the underlying Pf infection, the observed fever recall, and the relationship between the underlying infection and fever at that time. Results and Discussion should take this into consideration when interpreting results throughout the paper.

---

## [Author Response]

In particular, the reviewers and I agreed that you took an interesting approach to an important public health question, using a large, rich dataset. However we felt that there were methodological shortcomings in the model (or at least in the current description of it) that needed to be addressed. The main issues that need to be addressed are listed below.1) Model validationAlthough it is mentioned that the fever model has been validated (subsection “Model overview”), it is not clear how this was actually done. Part of the problem is that you have not presented any of the original data on fevers and malaria prevalence. This makes it impossible to tell the goodness of fit.Given that eLife has no space or figure restrictions, the following data plots would ideally be shown:- The raw 2x2 table showing all fevers versus RDT positivity (data and predictions), which is really interesting and is key to the message of the paper. This should also be plotted by country and transmission intensity.

A 2x2 table of RDT and fever positivity, from data and predictions, can now be found in an augmented Supplementary file 2. Additionally, we have added extra columns to Supplementary file 2 detailing the RDT/fever positivity breakdown by survey. The predicted national population-weighted prevalence of fever and RDT positivity can also be found in Supplementary file 1. Pie charts of national MAF, malaria-positive fever, and NMFI can also be found in Supplementary file 5.

A plot of the relationship between *Pf*PR_0-5_ and the fever metrics of interest (all-cause fever, malaria-positive fever, and malaria-attributable fever) at surveyed sites, and pixel-level predictions, can be found in Figure 5—figure supplement 1.

- Model-predicted vs. observed all cause fever by pixel or by country. A fairly small subset of the available DHS and MICS surveys are used, presumably because the rest do not have any data on RDT positivity (and/or GPS?). However these other surveys do contain data on all-cause fever and therefore would be an excellent source of validation data for this aspect of the model.

We used thirty-eight DHS and MICS surveys as both RDT result and GPS location were required for the model. Other DHS surveys do contain information on all-cause fever either at a national or ADMIN1 level collected typically over 3-4 months, often with data being collected in a non-randomised fashion (e.g. collecting data from one region at a time). Fever prevalence is likely to be highly seasonal and heterogeneous between (and within) countries due to the seasonality of diseases that contribute to all-cause fever.

For this reason, national or sub-national measurements of all-cause fever prevalence from these surveys cannot be directly compared with our estimates (or used directly to validate them), as we estimate the annual all-cause fever prevalence within an average two weeks in that year. Our model is able to bypass this restriction with georeferenced surveys by incorporating information on both the location and month of measurement within the model.

We do agree with the reviewers that these surveys are a valuable source of currently untapped information; a potential extension to this modelling framework would include predicting metrics at monthly time-steps to allow direct comparability with household surveys.

- A plot of the final fitted relationship between fever prevalence, parasite prevalence and the malaria-attributable fraction (and data on the 1st two variables)

These plots have been added as Figure 5—figure supplement 1.

- Comparison with observed vs. predicted fever prevalence. These data are not presented, and one of the reviewers who checked could not reconcile some of their predictions with the available data online. E.g. Results subsection “Prevalence of all-cause fever”: fever prevalence in Liberia is quoted as 51.7% in 2014, but on the DHS survey website only a Liberia survey in 2013, which gave fever prevalence as 28.6%. Similarly the most recent data from Niger (DHS 2012) shows a fever prevalence of 14.2%, but the quoted value by the authors is 57.2%. Please could you check and explain the reasons for any discrepancies.

For the same reasons explained in our second response to point 1, national surveys are not directly comparable to our predictions due to the differing temporal dynamics of the surveys and predictions, and the seasonality of all-cause fever.

Since improving the model with suggestions from the reviewers (described in detail in following answers), all the noted disparities have decreased. To investigate this fully, we extracted all-cause fever prevalence from 78 DHS/MICS surveys between 2006 and 2014; 72% of our predictions had a disparity of fewer than 10 percentage points from the household surveys. The majority of surveys with a disparity of more than 10 percentage points were from either the earlier or later years of the study period where less response data is available (the bulk of the surveys used were conducted in 2009-2012). As more data becomes available in later years, we expect model performance to improve.

- Related to those countries with the highest and lowest prevalence of fever (subsection “Prevalence of all-cause fever”): predicted values are given for Eritrea, Niger, Botswana and Zimbabwe. But in the supplementary data table (Supplementary file 2), the list of surveys does not include any data from any of these countries. Is this correct, or is the data list incomplete? If correct it seems concerning that the least certain estimates produce the outlying predictions. This should warrant some reinvestigation of the model fit. We suggest at least comparing the predictions against available fever prevalence estimates, even if not coupled with RDT data (as mentioned above).

The reviewers are correct that four out of the six countries we predict to have either the highest or lowest prevalence of all-cause fever are countries for which we do not have response data. In this exercise we have data from 38 household surveys, with each survey collecting data from one country across a few months. Our prediction spans 43 African countries across a 9-year time period, totalling 387 country-years. Our data therefore only covers a tiny fraction of this period so it is not unexpected that some of the highest and lowest predictions are in country-years with no data.

More generally, external validation should be performed by comparing model results to datasets not used to parameterise it. For instance you mention that case-control studies, transmission models etc. have been used to estimate the attributable fraction of malaria in fever – it would be useful to compare the model results with these studies.

We do discuss the contribution of case-control studies and transmission model estimates of MAF in the Introduction (subsection “Estimating malaria-attributable fever prevalence”).

We have overlaid the estimated clinical incidence from an ensemble of transmission models on our plot of all-cause fever, malaria-coincident fever, and malaria-attributable fever in Figure 5—figure supplement 1 (explained in more detail in our first response to point 1). As outlined in the Discussion (first paragraph), our MAF predictions match most closely with the transmission model estimates when the transmission models are predicting clinical incidence (i.e. fever) for the duration of the past 2-4 weeks (rather than the previous 0-2 weeks). The number of days prior to the interview since the onset of the child’s fever is known in 13 of the household surveys. In these surveys, the mean number of days since the onset of fever was 5.78 (SD: 3.85). Fevers in the interview are unlikely to always be detected in the final day of their natural cycle, so the close fit between our estimates and transmission model estimates of MAF within the past 2-4 weeks are comparable. The ensemble transmission model was calibrated using active case detection (ACD) studies where any fever occurring within the same 30-day period was counted as a single incident case of malaria (standard protocol for counting incident cases in ACD studies).

For transparency and to comply with eLife policy for mathematical models, you should provide the initial and final set of model equations (including coefficients) and the code used to select the final model, either in Supplementary Materials or in a suitable online repository such as github. Major data sets used that are not already in the public domain should also be provided unless there are compelling scientific or ethical reasons not to.

The model equations are given in the manuscript; we have added final model coefficients to Supplementary file 4.

The code for both the final multinomial model and the BRT covariate generation model are available at https://github.com/udalrymple/MAF-NMFI

2) Extending the model beyond environmental variablesAlthough in the fifth paragraph of the subsection “Covariates” it mentions that socio-demographic predictors of fever were considered, all the examples in the paragraph and in Supplementary file 3 are environmental. This seems like quite a large omission, especially given that fever is self-reported. As well as many varied causes of fever, cultural perceptions of fever are important here (and language, e.g. some languages do not have a specific word for fever). We were not clear how well environmental variables could be expected to predict this, especially given that you are extending fever predictions into countries for which there are no data.

We have revised the model to incorporate four indicators of national-level socio-demographic factors for 2006-2014, these are: percentage coverage of diphtheria, pertussis and tetanus (DPT) immunization; annual GDP growth; percentage of pregnant women who received prenatal care; and percentage with primary school education. These data were obtained from the World Bank Database and are described in full in the fifth paragraph of the subsection “Covariates” and Supplementary file 3.

We have also incorporated three additional socio-demographic variables at the same spatial resolution as the existing predictor variables (5x5km pixels) derived from the Global Urban Footprint dataset of global urban coverage; again these are described in full in the aforementioned paragraph and Supplementary file 3.

Additionally some of the variables used in the initial analysis are of a socio-demographic nature despite being remotely-sensed (accessibility, night-time lights).

Besides cultural factors, a list of suitable covariates may include socioeconomic factors such as nutrition, crowding, mother's education, access to clean water etc. Many of these are available in DHS/MICS data but it is difficult to see how they were used. Indeed, given that the final model contained 167 predictor variables, it is difficult to see why there was any human selection at all. The authors have not tried to justify any of their chosen variables based on causality arguments, so would it not make more sense to use the entire DHS dataset as predictors, and then let variable selection algorithms winnow this down?

The predictor variables used in the creation of the covariate surfaces (*Pf*PR_0-5_ and the BRT predicted surface of background fever prevalence) are required to be complete surfaces of known values across the continent, prompting the decision to use remotely-sensed variables. These variables (with the exception of the variables added in the revision, described in the answer to Q4) were the same as those used in the MAP *Pf*PR_2-10_ model (Bhatt et al., Nature 2015), with the logic that this broad suite of variables are inclusive of most currently available geospatial data informing on the distribution of diseases that are constrained by environmental conditions.

We agree with the notion that other DHS variables would be suitable predictors of MAF and NMFI prevalence; however their values are unknown in locations other than the survey sites and so cannot function as covariates in a predictive spatial model. We have addressed this in more detail in our next response.

We are also concerned about the apparent lack of variables at the household or even individual level. This may be a problem if being malaria-positive is (positively or negatively) associated with having a fever beyond what can be explained by the spatial covariates examined. For instance, within a particular town or village (with homogeneous environmental variables) there will be poorer households who are more likely to be both malaria-positive and to have non-malaria fevers. Within the household there will be further associations in distribution due to the age, gender, birth order, genetic makeup etc. of individuals. Perhaps this has been taken into account, but it is not clear how this happened.

Throughout this study we have placed greater emphasis on predicting the patterns of fever throughout Africa and relative contributions of malaria and other causes. As such we have configured our model to predict these quantities for 5x5km pixels across Africa. As in the previous response, this precluded the use in our predictive model of any individual-level or household-level covariates since these are not available on a pixel-by-pixel basis. We agree with the reviewers that at the individual level many survey covariates (such as housing type and income level) are likely to be significantly predictive of both malarial and non-malarial fever prevalence, and moreover able to explain a correlation of increasing risk for both. Including this information in our model could potentially have increased predictive power for the former effect and reduce the possibility of over-dispersed errors contributed by the second effect. However, we have not included these individual level predictor variables in our current model owing to the heterogeneity of individual level variables measured in different surveys and the lack of a common set of information directly representative of those variables outside of surveyed countries.

During model building we attempted fitting an over-dispersed multinomial model but encountered a number of numerical difficulties suggestive of an ill-conditioned Laplace approximation step. Hence we remain concerned as to the potential for some errors due to ignoring the potential for correlated risk factors within cells. We have made a new note of this in our discussion and draw some reassurance from the probability integral transform diagnostic (Figure 1—figure supplement 1) that suggests our model uncertainties are reasonably well, though not perfectly, calibrated. In order to further explore the potential extent of this effect we have examined the relationship between leave-one-out predictive error for *Pf*PR and NMFI and the standard deviation of relative wealth (a variable consistently reported across surveys), and found no significant effect when plotted against the PIT histogram or with quantile regression at quantiles 0.025, 0.5 and 0.975.

In general, we get the impression that the ecology of the pathogen, its environment and insect host is well-described, but the epidemiological, immunological, cultural and socioeconomic determinants of disease within the human host are either less well captured or at least less well explained. Perhaps you should include someone with clinical or at least public health training in your authorship list.

Following our last two responses, we reiterate that our focus has been on continental prediction of the quantities of interest and their interaction rather than to explicitly offer new insight into the contextual factors listed above. We agree these are interesting and worthy of further investigation but propose that doing so is not an essential component of the current study.

3) ScopeThere are a number of areas where we believe that the scope and implications of the results may be overstated.- You need to clarify that this study aims at improving burden estimates of uncomplicated malaria, and not case management policy. WHO clearly recommends, as do all national policies in African countries with endemic malaria, that all fevers/suspected malaria presenting at facilitates in malaria endemic areas should receive a laboratory diagnosis for malaria, and if positive treated with the first-line antimalarial. This doesn't mean the attending health professional cannot go on to treat other presenting illnesses and symptoms. But even if the fever is not directly attributable to the Pf infection at that time, it should be treated. This needs to be made clear in the paper. You should stick to how these findings impact the overall epidemiology of fever illness among children in Africa, and not make recommendations or draw conclusions from this study in the discussion for malaria case management (or IMCI) policy.

One of the major findings of this paper is that many fevers that are coincident with a malaria infection are causally attributable to another disease. This analysis is the first, to our knowledge, to quantify this; and the spatial heterogeneity in the proportion of fevers attributable to *P. falciparum* malaria and other diseases means that a febrile child has a differing chance of having MAF or NMFI depending on where they are in Africa. This does have ramifications on the likelihood of missing a co-infecting disease given a positive RDT at the clinic, should the child be infected with a disease with overlapping symptoms to *P. falciparum* malaria.

IMCI for fever in high to low malaria risk, in situations when the child’s RDT tests positive, recommends looking also for a bacterial cause of fever from external examination of temperature, stiffness, lesions etc. However if no external symptoms are present an antibiotic will not necessarily be provided. The child will, at the end of the consultation, receive a first-line antimalarial and an antibiotic if the visual examination suggests that they have a bacterial co-infection. By claiming that current WHO guidelines for integrated management of childhood illness (IMCI) currently suboptimal for the treatment of fever, we intend to convey that the diagnosis of NMFI is far less robust than for malaria, given that it is a visual diagnosis rather than a parasite based diagnosis, and could be improved should diagnostic tests for common NMFIs become more widely available.

We have adapted the Discussion to convey this more clearly.

- You argue that the results of their work will improve burden estimates. We find this to be somewhat of a 'straw man' attack, as to our knowledge no burden estimates have been based on an RDT positive child with a history of fever in the past 2 weeks. Neither WHO GMP, MAP nor GBD uses such a method.

The reason for using RDT positivity, rather than microscopy, is the advantage of HRP2 persistence giving a more accurate indication of two-week positivity. We are unable to use microscopy results (although available from the household surveys) as microscopy results are not a reliable indicator of 2-week malaria positivity in situations where an individual has already received antimalarial treatment for their fever. Additionally, as RDTs become more widely available, they may be used in the future for burden estimates, particularly in new 'real-time' burden estimations from new routine case reporting systems (such as DHIS2).

When estimating the total malaria burden, confirmed malaria “cases”- that is, a febrile individual who seeks care at a facility that reports to a national HMIS and receives a positive *P. falciparum* diagnosis- are utilised to calculate the total incidence of malaria nationally. If many of these malaria cases are actually asymptomatic malaria cases with a coincident and symptomatic NMFI, then the number of these types of malaria “case” would vary due to the prevalence of NMFI rather than the prevalence of malaria. A more accurate (although clinically challenging) measurement of malaria cases would be to calculate the total number of malaria-attributable fevers, rather than those that simply present at a clinic with a positive RDT. We would expect that, in the hypothetical complete absence of NMFI, the number of individuals presenting with a fever at clinics would decline and subsequently the total number of positive RDTs within clinics would also decline.

Additionally, the treatment-seeking rate for all-cause fever is used as a further adjustment to calculate national *P. falciparum* incidence. It is possible that the treatment seeking rate for MAF and NMFI is not equivalent. While measuring the treatment seeking rate for MAF and NMFI is not within the scope of this paper, a better understanding of the causal contributions of MAF and NMFI to all fevers is crucial to enhancing our estimates of malaria burden in the future.

- You present cross-sectional household survey data that measures a 2-week (or there about) RDT period prevalence based on persisting HRP2 antigenemia from a Pf infection, plus an overlapping fever history based on the recall by the mother/caregiver. Their primary results suggest a large proportion of these fevers are not directly attributed to the underlying Pf infection. While this seems an appropriate interpretation of the results and in line with malaria epidemiology, the cross-sectional nature of the study is a major limitation. You need to make note that the underlying Pf infection likely would have resulted in at least 1 parasite-attributable fever, likely in the first month of the infection, and additional parasite-related fevers will likely occur, especially if a new infection occurs on top of the existing infection (just based on the malaria therapy data). So the timing of the observed RDT+ and fever is important in understanding the true relationship between the underlying Pf infection, the observed fever recall, and the relationship between the underlying infection and fever at that time. Results and Discussion should take this into consideration when interpreting results throughout the paper.

We accept these comments and have adjusted the Results and Discussion accordingly. “…[I]n any given two-week period” has been added, and we also added a discussion point on ever-symptomatic fevers in the Discussion (third paragraph).